# Cardiac Arrhythmias in Patients Treated for Lung Cancer: A Review

**DOI:** 10.3390/cancers15245723

**Published:** 2023-12-06

**Authors:** Maja Hawryszko, Grzegorz Sławiński, Bartłomiej Tomasik, Ewa Lewicka

**Affiliations:** 1Department of Cardiology and Heart Electrotherapy, Faculty of Medicine, Medical University of Gdansk, Smoluchowskiego 17 Street, 80-214 Gdansk, Poland; maja.klimkiewicz@gmail.com (M.H.); elew@gumed.edu.pl (E.L.); 2Department of Oncology and Radiotherapy, Faculty of Medicine, Medical University of Gdansk, Smoluchowskiego 17 Street, 80-214 Gdansk, Poland; bartlomiej.tomasik@gumed.edu.pl

**Keywords:** lung cancer, cardio-oncology, cardiac arrhythmias, radiation therapy, stereotactic arrhythmia radioablation

## Abstract

**Simple Summary:**

Anticancer treatment can lead to cardiotoxicity, including arrhythmias, which are the least understood complication. Atrial fibrillation is common in this patient population, and cancer patients may be at increased risk for thromboembolic events. Ventricular arrhythmias are less common than supraventricular arrhythmias. Diagnostic methods are the same as in patients without cancer, i.e. ECG, Holter monitoring, electrophysiological study and echocardiography. Amiodarone is recommended for supraventricular and ventricular arrhythmias. There are also possible invasive treatments for arrhythmias - transcatheter ablation and stereotactic radioablation of arrhythmias - but these are used much less frequently than drug treatment. There is limited data on atrioventricular conduction abnormalities in lung cancer, probably related to immune checkpoint inhibitor-induced myocarditis. Cancer and heart disease are leading causes of death, underscoring the importance of developing cardio-oncology and training specialists to treat cardiovascular complications in cancer patients.

**Abstract:**

Cardio-oncology currently faces one of the greatest challenges in the field of health care. The main goal of this discipline is to ensure that patients treated for cancer do not suffer or die from cardiovascular disease. The number of studies on the mechanisms of heart injury during cancer treatment is constantly increasing. However, there is insufficient data on heart rhythm disorders that may result from this treatment. This issue seems to be particularly important in patients with lung cancer, in whom anticancer therapy, especially radiotherapy, may contribute to the onset of cardiac arrhythmias. The observed relationship between cardiac dosimetry and radiotherapy-induced cardiotoxicity in lung cancer treatment may explain the increased mortality from cardiovascular causes in patients after chest irradiation. Further research is essential to elucidate the role of cardiac arrhythmias in this context. Conversely, recent reports have highlighted the application of stereotactic arrhythmia radioablation (STAR) in the treatment of ventricular tachycardia. This review of available studies on the epidemiology, pathogenesis, diagnosis, and treatment of arrhythmias in patients treated for lung cancer aims to draw attention to the need for regular cardiological monitoring in this group of patients. Improving cardiac care for patients with lung cancer has the potential to enhance their overall therapeutic outcomes.

## 1. Introduction

Lung cancer (LC) is still one of the leading causes of cancer-related death worldwide for both men and women [1]. There are two main histopathological types of LC, which differ clinically: non-small cell lung cancer (NSCLC, 80–85% of lung cancer patients) and small cell lung cancer (SCLC, 10–15%) [2]. Importantly, more than half of NSCLC patients have metastases at the time of diagnosis [2].

Over the last decade, the use of advanced chemotherapy, radiotherapy (RT), targeted therapies, and immunotherapy has significantly improved outcomes and prognosis in this group of patients [3,4]. However, anti-cancer treatment may have adverse effects on the cardiovascular system [5]. Approximately 75% of LC patients are considered suitable candidates for RT, which can be used in both curative and palliative setting [6]. While advances in conformal techniques have allowed the beam to target more accurately tumor volume, surrounding healthy tissues continue to receive radiation doses leading to radiotherapy-induced toxicities [7]. Cardiac arrhythmia has been identified as a potential negative outcome of unintentional exposure to radiation during the treatment of LC [8,9,10]. Atrial tachyarrhythmias, such as atrial fibrillation (AF), are the most common rhythm disturbances observed in general cardio-oncology clinics [11,12].

Cardiac arrhythmias, in the context of cardiotoxicity of the applied oncological treatment are difficult to assess due to the number of potential factors influencing their occurrence. LC patients constitute a heterogeneous group and include people who were previously considered healthy, as well as patients with multiple comorbidities, including those with cardiac diseases or arrhythmias diagnosed before the initiation of cancer treatment [13].

Arrhythmias may be asymptomatic or worsen the patient’s quality of life and general condition, and even lead to sudden cardiac death (SCD). Due to the high incidence of LC, and improved prognosis in these patients, the problem of cardiotoxicity, including cardiac arrhythmias, is an increasingly important clinical problem [14].

## 2. Pathogenesis of Cardiac Arrhythmias in Patients Treated for Lung Cancer

Generally speaking, supraventricular and ventricular arrhythmias, atrioventricular blocks, and sinus bradycardia are among the less common side effects of anticancer therapy [15]. This may be due to the lack of robust prospective studies on cardiac arrhythmias in cancer patients, including those treated for lung cancer. However, some antineoplastic drugs are well known as having a direct or indirect arrhythmogenic potential. The cause of arrhythmias is still not entirely clear, but growing evidence supports the interrelation of various mechanisms involved in anticancer drug damage to the heart muscle. These probably include changes in the membrane activity of ion channels, disturbances in calcium homeostasis in cardiomyocytes resulting in changes in their electrophysiological properties (anthracyclines, taxanes, antimetabolites, alkylating drugs), and influence on the function of ion channels and cellular signaling pathways (tyrosine kinase inhibitors) [12]. In patients treated with immunotherapy, arrhythmias occur mainly in the course of myocarditis induced by these drugs; however, the incidence of these events is low [16]. Arrhythmias that may occur during chemotherapy, targeted therapies, or immunotherapy for LC are shown in Table 1.

RT is an integral part of LC treatment. The processes involved in radiation-induced cardiac toxicity are multifaceted and not fully comprehended, encompassing oxidative stress, DNA damage, and the activation of inflammatory and profibrotic cytokines. These mechanisms ultimately lead to the injury and formation of fibrosis within critical components of the heart such as myocardium, coronary arteries, valves, and pericardium [27]. Unlike chemotherapy, targeted therapies, or immunotherapy, injury is usually recognized later after exposure. However, it has been reported that a significant proportion of patients may develop arrhythmias within months after treatment [9].

Cardiac complications are related not only to the cancer itself and its treatment, but also to comorbidities, other non-oncological drugs, and electrolyte disturbances resulting from frequent side effects, such as vomiting or diarrhea, accompanying oncological treatment. Therefore, close monitoring of serum potassium, magnesium and calcium levels is recommended in this group of patients. Even small deviations from the norm in terms of dyselectrolytemia should also be corrected, which allows for more effective prevention and treatment of, for instance, paroxysmal atrial fibrillation [28,29,30].

In cancer survivors, arrhythmias may occur many years after completing oncological treatment. In the course of dilated cardiomyopathy developing 5–10 years after cancer treatment, mainly with anthracyclines, recurrent, clinically malignant arrhythmias (sustained ventricular tachycardia, ventricular fibrillation) may occur, which may cause SCD. 

## 3. Diagnostics

Diagnosis of arrhythmias in cancer patients is based on standard diagnostic methods, such as 12-lead electrocardiogram (ECG) or Holter ECG monitoring. The use of mobile health (mHealth) devices, such as wearables and health tracking applications, is becoming increasingly popular, especially in the case of short-term lasting arrhythmias. It usually allows a diagnosis to be made and the appropriate treatment to be initiated [31]. Previously, the idea of using smartwatches was to detect AF, and it has been shown that smartwatch-based single-lead ECG appears to be a reasonable alternative to standard Holter monitoring [32]. Currently, there is much data on the diagnosis of other arrhythmias thanks to these devices, both supraventricular and ventricular, as well as atrioventricular conduction disturbances or sinus bradycardia [33]. In very rare cases, in the absence of confirmation of arrhythmias by non-invasive tests, electrophysiological study (EPS) may be considered [31].

Transthoracic echocardiography (TTE) is essential in cancer patients with cardiac arrhythmias to recognize concomitant heart disease or to assess the dimensions of the left atrium (LA) in a patient with AF before deciding to restore sinus rhythm. Transesophageal echocardiography (TEE) or cardiac magnetic resonance imaging (cMR) may be required when there is a suspicion of cardiac metastasis or infiltration by cancer. TEE or cardiac computed tomography (CT) allow the exclusion of the LA appendage (LAA) thrombus before electrical cardioversion for AF. In patients with exercise-induced arrhythmia a treadmill or cycle ergometer exercise test may be necessary to exclude ischemia.

## 4. Supraventricular Arrhythmias

### 4.1. Atrial Fibrillation

AF is the most common arrhythmia in the general population (2–4%), and its incidence increases with age [14]. Caucasian men are more commonly affected. Other contributing factors include hypertension, diabetes, hypothyroidism, ischemic heart disease, chronic heart failure, valvular diseases, obesity, sleep apnea syndrome, and the use of stimulants [14].

AF is much more common in cancer patients than in the general population (2–16%), and the highest incidence is reported in those after LC surgery—ranging from 6% to 32% [34,35,36]. Imperatori et al. showed that almost 10% of patients after lobectomy had postoperative AF [36]. Many of these patients were elderly and had cardiac disease. Interestingly, patients with left-sided lung cancer were affected more often than those with right-sided LC. Patients with AF had longer hospital stays, with higher intensive care unit admission rates and in-hospital mortality. It was shown that postoperative AF is a predictor of poor long-term survival in this group of patients.

There are various factors affecting the incidence of AF in LC patients. This may be related to comorbidities, direct effects of the tumor, or complications of medical and surgical treatment, with inflammation being the most common denominator in all of these cases. AF may also be the first symptom of LC metastases located within the pulmonary veins or in the LA [37,38].

Cancer patients with AF have significantly increased thromboembolic risk, including a 2-fold higher risk of stroke. They also present with a 6-fold higher risk of heart failure, and a 2-fold higher mortality rate. Therefore, AF is an important factor influencing the prognosis during lung cancer treatment and is a challenge in therapeutic management [14,39]. The use of CHA2DS2-VASc score (Table 2A) has been proposed to assess the risk of stroke/systemic embolism in cancer patients with AF, as recommended by current cardio-oncology guidelines [14]. However, this scoring is not intended to identify high-risk patients, but rather low-risk individuals who may avoid anticoagulant therapy. Moreover, the CHA2DS2-VASc score has not been validated in cancer patients with AF and may underestimate the risk of thromboembolic complications in this group of patients.

Despite the imperfections of the CHA2DS2-VASc score, the latest European Society of Cardiology (ESC) guidelines on cardio-oncology recommend initiating anticoagulant therapy in people with a score ≥ 2 in men and ≥ 3 in women (class I C recommendation) and considering anticoagulation when the score is 1 for men and 2 for women (class IIa C). In addition, cancer patients with AF and a score of 0 in men and 1 in women may actually have a higher thrombotic risk, so anticoagulation can be considered in these patients after considering the risk of bleeding (class IIb C). The HAS-BLED score (Table 2B) may be useful in assessing bleeding risk; however, in lung cancer patients the presence of hemoptysis or infiltration of large vessels by the tumor should be taken into account. Once a decision has been made to apply anticoagulation, thromboembolic and bleeding risk should be reassessed during follow-up (class I C class) [14].

In LC patients with AF, the challenge is not only to decide to start antithrombotic therapy, but also to choose an anticoagulant. In patients with moderate to severe mitral valve stenosis or a mechanical prosthetic valve, vitamin K antagonists (VKAs) remain the only antithrombotic option.

Low molecular weight heparins (LMWHs) can be used for short-term anticoagulant treatment in hospitalized patients with a recent diagnosis of cancer and during some anticancer therapies. However, it needs underlying, that even in the general population the efficacy of LMWHs in the prevention of stroke or systemic embolism in AF patients has not been evaluated in any randomized clinical trial. Therefore, according to the guidelines, LMWHs, as well as antiplatelet therapy are not recommended in cancer patients with AF (Class III C recommendation).

Among the therapeutic options are non-vitamin K antagonist oral anticoagulants (NOACs), but these have been evaluated in randomized clinical studies in patients with cancer and venous thromboembolism (VTE), but not with AF. However, post hoc analyses of NOAC trials with direct factor Xa inhibitors (ROCKET AF, ARISTOTLE, ENGAGE AF-TIMI) and observational data suggest greater safety and at least similar efficacy of NOACs compared to VKAs in patients with AF and active cancer, after excluding a high bleeding risk, significant drug-drug interactions, or severe renal impairment (Class IIa B recommendation in recent guidelines). However, in patients with active cancer and AF in whom NOACs cannot be used, LMWH should be applied (Class IIa C) [14].

If long-term anticoagulation is contraindicated or there is a high risk of bleeding (for instance HAS-BLED score > 4), LAA closure should be considered in patients with a life expectancy > 12 months (Class IIb C) [14]. The latest data suggest that such treatment, as in the case of patients without a diagnosis of malignancy, is associated with good procedural success and has offered a reduction in stroke at no increased bleeding risk [40]. Importantly, also the long-term results of such treatment in the group of patients with cancer do not differ compared to patients without cancer in terms of mortality, stroke, and major bleeding episodes [41].

### 4.2. Atrial Flutter and Atrial Tachycardia

Atrial arrhythmias may occur in patients treated with alkylating agents (cisplatin, cyclophosphamide, ifosfamide, melphalan), amsacrine, anthracyclines, antimetabolites (capecitabine, 5-FU, methotrexate), bortezomib, doxorubicin, IL-2, interferons, paclitaxel, ponatinib, and romidepsin [15]. However, data on their incidence are scarce.

These arrhythmias can be treated with antiarrhythmic drugs and interruption of anticancer therapy is rarely needed. In antiarrhythmic treatment amiodarone should be preferred, as the most effective antiarrhythmic drug (it is recommended to check thyroid hormones before using it).

Atrial flutter (AFL) is associated with a similar risk of thromboembolic complications as with AF, so the management regarding antithrombotic treatment is the same.

## 5. Management of Supraventricular Arrhythmias in Lung Cancer Patients

Because AF is common in LC patients in the perioperative period, a number of studies have examined the use of antiarrhythmic drugs to prevent its occurrence. Some studies have shown that intravenous amiodarone, followed by oral administration for several days, is effective and safe in patients undergoing lung cancer surgery [42,43,44]. The addition of N-acetylcysteine to amiodarone had no additional beneficial effect [45]. There are also data on the use of losartan, beta-blockers, magnesium sulfate, or calcium-channel blockers in the prevention of perioperative AF. According to the literature, prophylaxis with beta-blockers seems to be the most effective and safest of the treatments studied [46,47].

Data on the effectiveness of antiarrhythmic drugs (AADs) in preventing AF recurrence in patients with lung cancer are lacking. Interactions between targeted therapies and AADs can significantly impact the treatment strategy for AF. AADs may result in inhibition of P-glycoprotein-mediated transport of the targeted drug or increase concentrations of one or both drugs, often due to the impaired hepatic cytochrome P450 metabolism. An increased propensity for QT interval prolongation and bradycardia has also been observed. Therefore, some AADs should be used with careful ECG monitoring and/or appropriate dose adjustment, while others are contraindicated in combination with certain targeted therapies. Drug–drug interactions may represent a major challenge in the management of AF in case of active malignancy and constitute an important topic for further research [48].

Asnani et al. examined the effects of AADs when used with targeted therapies, most of which were based on tyrosine kinase inhibitors. Class IA, IC, and class III antiarrhythmic drugs most commonly cause QT interval prolongation when used in combination with targeted therapies. Amiodarone, dronedarone, and calcium channel blockers can interact with almost all targeted therapies, and AAD dose adjustment is often required or to be discontinued. In contrast, class IB antiarrhythmics, particularly mexiletine, are the least likely to cause significant drug–drug interactions. Among class II drugs, carvedilol, propranolol, and nadolol are more likely to be involved in drug–drug interactions than metoprolol, atenolol, and pindolol. Apart from their effects on targeted therapies, their use may be limited by bradycardia or low blood pressure [48].

Catheter ablation (CA) is a well-established therapeutic option for the treatment of symptomatic, predominantly paroxysmal, and drug-refractory AF. However, in clinical trials on CA for AF, the diagnosis of active cancer was an exclusion criterion from these studies [49]. There is limited data evaluating the effectiveness and safety of CA for AF in patients with active cancer coming mainly from case reports. Tamura et al. reported on successful RF ablation performed in a patient with a drug-refractory atrial tachycardia originating from a pulmonary vein invaded by lung cancer [50]. More data have come from studies evaluating results of CA in cancer survivors.

Giustozzi et al. evaluated the safety of CA in the treatment of non-valvular AF in patients with an oncologic burden. These were mainly patients with a history of gastrointestinal, breast, and genitourinary cancers. Clinically significant bleeding was more common in cancer survivors than in patients without cancer. Importantly, the type of anticoagulation was not associated with bleeding [51]. However, in another retrospective study, CA for atrial fibrillation was shown to be a safe and effective treatment in patients with a history of cancer and in those exposed to potentially cardiotoxic anthracycline and/or thoracic radiotherapy [52]. In this study, 251 patients were in the cancer group, including 15 (6%) patients with lung cancer, and 46 patients (18.3%) who were undergoing anticancer treatment at the time of CA. At 12 months after CA, freedom from AF did not differ among patients with and without cancer (83.3% vs. 72.5%, *p* = 0.28). Outcomes for possible complications, such as postprocedural bleeding, cardiac tamponade, stroke, and pulmonary vein stenosis during the first 3 months after CA were similar to those obtained in the non-cancer control group. A likely reason for the excess bleeding risk observed by Giustozzi et al. was their practice of bridging with LMWH after CA, rather than continuing anticoagulant treatment without interruption, which is currently recommended [51]. According to Ganatra et al. there are no differences in the safety of CA due to AF in cancer survivors, and a history of cancer or cancer-related therapy is not associated with the risk of AF recurrence after CA, as shown by multivariate regression analysis [52].

## 6. Ventricular Arrhythmias

Ventricular arrhythmias (VA), although much less common than supraventricular arrhythmias, can complicate anticancer therapies and lead to serious adverse events, including SCD. Their pathogenesis includes direct and indirect effects of oncological treatment on cardiomyocytes, direct injury or myocardial ischemia, systemic inflammation, electrolyte disturbances, QT interval prolongation, as well as the effect on the autonomic nervous system [12,53]. Like AF, VA may be the first symptom of lung cancer infiltrating the heart [54]. Prevention of VA in cancer patients focuses mainly on modifiable risk factors, including correcting electrolyte disturbances and avoiding concomitant use of drugs that may increase the potential for arrhythmia [12]. An ECG should also be performed to document the baseline QTc interval for later reference during follow-up visits. There is no risk scale to stratify an individual patient’s risk of QTc prolongation during anticancer therapy, therefore regular ECG monitoring is essential [55]. It is recommended to measure the QT interval in leads II or V5 (without the U wave), averaging 3–5 measurements during sinus rhythm and about 10 measurements during AF. When measuring the QTc interval in cancer patients, it is useful to use the Fridericia formula (QTcF = QT/∛RR), because at heart rates > 100 bpm and < 60 bpm, the QTc obtained from Bazett’s formula (QTcB = QT/√RR) may be underestimated [55].

## 7. Management of Ventricular Arrhythmias in Lung Cancer Patients

Considerations regarding the use of AADs to prevent VA recurrence in patients with lung cancer are the same as those for AF and other supraventricular tachyarrhythmias. If there is no risk of serious interactions with the oncological drugs, the use of amiodarone, which is the most effective antiarrhythmic drug, is the most reasonable.

In patients with structural heart disease, AADs and implantable cardioverter-defibrillators (ICDs) are used to minimize the risk of SCD from ventricular tachycardia (VT) or ventricular fibrillation (VF). Patients with recurrent VT and refractory to AADs are referred for minimally invasive CA. However, many patients with advanced heart disease have comorbidities that make invasive procedures difficult or impossible. Prior cardiac procedures or other conditions, including cancer, can be a limitation to CA, but this is the localization of the arrhythmic substrate for VT which commonly determines CA efficacy. In some patients with recurrent VT and ineffective CA, or when CA is not possible, cardiac stereotactic body radiation therapy (SBRT) was beneficial [56,57,58]. If it is used in antiarrhythmic treatment for VT it is then called stereotactic arrhythmia radioablation (STAR). The underlying mechanism responsible for the rapid reduction in VT frequency and magnitude remains unclear, challenging the previously proposed theory of radiation-induced fibrosis [59,60]. Patient eligibility for this procedure remains controversial [61]. Results from some studies have shown that STAR is associated with a significant reduction in the burden of ventricular tachyarrhythmias and ICD interventions, particularly during the first months after the procedure. Meta-analysis of seven observational studies with a total of 61 patients showed that six months after STAR, the VT burden reduction was 92%, accompanied with an 86% reduction in the number of ICD shocks [62]. Several small prospective trials were recently published [63,64,65,66,67]. These studies primarily focused on safety and indicated that STAR has acceptable short-term toxicity with no significant concerns regarding acute cardiac toxicity. However, in most cases, cardiac SBRT did not result in complete resolution of VT in long-term follow-up. Therefore, it is currently recommended to use cardiac SBRT in critically ill patients with treatment-resistant (including CA) ventricular tachycardia and electrical storm [68].

A large research network for investigating STAR results in the treatment for VT was recently established with the aim to evaluate patterns of practice and outcomes of STAR and finally to harmonize STAR within Europe [56].

## 8. Implantable Cardioverter-Defibrillator (ICD)

ICD implantation is an established treatment for malignant ventricular tachyarrhythmias. However, there are few studies evaluating the effect of cancer on the severity of VA in patients with ICD and most of them are retrospective. Enriquez et al. analyzed 1.598 patients, of whom 209 (13.1%) had a diagnosis of cancer (9% of them had lung cancer) and 102 patients (6.4%) were diagnosed with cancer after ICD implantation (14% of them with lung cancer). Over the 7 years of the study, 89 out of 204 cancer patients experienced at least one episode of VT or VF (43.6%) [69]. After ICD implantation, the mean time to cancer diagnosis was 41.1 ± 28.1 months. VA occurred in 32% of patients after cancer recognition, and the arrhythmic burden increased significantly compared to the period before the diagnosis of cancer. The incidence of VT/VF was significantly higher in patients with stage IV cancer than in those with less advanced disease [70].

Christensen et al. analyzed 5665 patients treated with ICD, 5.1% of whom had previously been diagnosed with cancer. LC patients constituted 7% of those who were implanted with an ICD for primary prevention of SCD, as well as 12% of cancer patients implanted with an ICD for secondary prevention [71]. Cancer was associated with a significantly higher risk of death in the first year and beyond in patients with ICD implanted for secondary prevention of SCD. In another study, there were no differences in appropriate ICD therapies between patients with or without cancer, regardless of ICD indication [71].

When qualifying a cancer patient for treatment with an ICD, a cooperation with an oncologist is necessary, because the required criterion is the expected survival of more than 1 year. It should be considered that a routine transvenous ICD lead implantation may be difficult due to axillary lymph node dissection or the presence of other vascular access devices. In such cases the use of a wearable cardioverter-defibrillator (WCD) may be considered, or, in exceptional cases, implantation of a subcutaneous ICD (S-ICD) [11]. It should be noted that an ICD is not a contraindication of RT for LC. Monitoring of the patient during irradiation is required, but no disturbances in the ICD functioning have been observed [72,73].

## 9. Atrio-Ventricular Conduction Disturbances and Sinus Bradycardia

Data on atrioventricular conduction (AV) disorders in lung cancer patients are scarce, probably due to their rarity. AV disturbances, also in the form of complete AV block, may be a manifestation of fulminant myocarditis complicating the treatment with immune checkpoint inhibitors (ICIs) [74,75]. In the case of advanced AV conduction disorders in the course of ICI-related myocarditis, the use of temporary cardiac pacing and the administration of methylprednisolone is of crucial importance [76].

There are case reports on AV conduction disturbances secondary to LC metastases to the heart [77,78]. Also, endobronchial brachytherapy due to lung cancer may be associated with newly developed intraventricular conduction disorders in the form of left bundle branch block (LBBB). These may be transient but may also be associated with symptoms of acute heart failure [79].

Severe and persistent sinus bradycardia is also rarely observed during treatment of LC patients. It should be emphasized that most drugs used for LC treatment can only cause transient bradycardia, which disappears after treatment and pacemaker implantation is not necessary.

Alectinib used for an ALK-positive NSCLC is known to induce sinus bradycardia. In a study, that included 53 patients treated with alectinib, a median reduction in heart rate of 17 bpm was noted, resulting in bradycardia in 42% of patients. However, it was symptomatic in six patients and only one patient required pacemaker implantation [80].

## 10. Lung Cancer Treatment

### 10.1. Surgery

Surgical procedures can be performed using standard access—thoracotomy or via thoracoscopy. Thoracoscopic methods are preferred due to their minimal invasiveness, faster patient recovery, reduction of postoperative pain, and shorter hospital stay [81,82].

However, a significant relationship was observed between the surgical treatment of cancer and the subsequent occurrence of arrhythmias. AF and other supraventricular arrhythmias are among the most common complications after thoracic surgery, especially after lung resection. They were reported in 38.1% of patients after pneumonectomy [83]. It has been shown that the occurrence of arrhythmias is associated significantly with an increase in postoperative morbidity and mortality. When searching for risk factors for postoperative supraventricular arrhythmias, many variables were found. These include accompanying chronic diseases, weight loss, history of smoking, time to smoking cessation, hypertension, congestive heart failure, angina, previously documented arrhythmias, peripheral vascular disease, chronic obstructive pulmonary disease (COPD), previous thoracotomy, mediastinal surgery, and intraoperative blood transfusion [84,85]. However, the most studied and recognized as the main risk factors for AF after LC surgery are older age, male gender, and the size of lung resection [84,85].

### 10.2. Radiotherapy

Radiotherapy for LC is used for curative or palliative purpose, alone or in combination with chemotherapy. The risk of cardiac complications in lung cancer patients increases with the total dose of radiation, the volume of the heart exposed, and the use of chemotherapy [27,86]. After the publication of the results of the seminal RTOG 0617 trial, a follow-up analysis conducted two years later revealed that the volume of the heart receiving 40 Gy or more (V40) strongly correlated with patient survival [87]. Recently, the dose to critical cardiac substructures, has been systematically studied in LC [88]. With the improvement of heart-sparing techniques, the incidence of radiation-induced cardiac dysfunction has decreased but still remains a serious problem [27]. It is postulated that the cardiotoxicity of RT used in the treatment of LC, which may result in poorer overall survival, is underestimated. The arrhythmia substrate may arise through the mechanism of radiation damage to the coronary arteries, resulting in myocardial ischemia and ultimately myocardial fibrosis. The mechanisms of radiation-induced cardiac arrhythmias are presented in Figure 1. Radiotherapy is associated with various non-specific ECG changes, including QTc interval prolongation, T-wave changes, and reduced QRS complex amplitude, but their relevance to the occurrence of arrhythmias is not documented [89]. The pathophysiology of atrial fibrillation induced by radiotherapy is most likely based on inflammation and later fibrosis of the atrial wall involved in the radiating field. Moreover, RT affects the autonomic system, leading to inappropriate sinus tachycardia and to reduced heart rate variability [90,91,92,93].

Data on cardiac arrhythmias due to RT are scarce. In a retrospective study, which included 76 LC patients treated from 2010 to 2015 with conventionally fractionated RT, atrial arrhythmias were reported in five patients [68]. The long-term effects of radiotherapy, such as coronary artery disease, cardiomyopathy, valvular heart disease, and heart failure, are much better understood (Appendix A) [4]. However, most data come from patients with hematological or breast cancer, and there is a lack of data from LC patients. This is undoubtedly due to their low survival rate. However, in recent years it has been documented that RT-related cardiotoxicity may also have an impact on all-cause mortality in lung cancer patients in the first months and years after this treatment [94].

In a retrospective study on 11,455 patients with NSCLC, Mo et al. found that over 3 years of follow-up, mortality due to LC decreased, while mortality due to cardiovascular diseases associated with RT increased significantly [89]. Kim et al., assessing 560 LC patients treated with RT, found that incidental irradiation of the sinoatrial node may be associated with the subsequent development of AF and increased mortality [95].

It was also recognized that the risk of arrhythmia depends on the dose of irradiation applied to a given area in the heart. It was shown in lung cancer patients that V5Gy ≥ 60 cc for the left atrium was associated with an increased risk of AF, and V60Gy ≥ 0.03 cc for the right atrium increased the risk of other supraventricular tachycardia (SVT). The dose V5Gy ≥ 1 cc for the left main coronary artery was a risk factor of VT occurrence, and V10Gy ≥ 1 cc increased the risk of bradyarrhythmias [96]. There is now increasing evidence of an association between cardiac dosimetry and RT-related cardiotoxicity in patients irradiated for lung cancer. The base of the heart was identified as the area most sensitive to radiation damage. This anatomical area includes the atria, sinus node, and proximal segments of the coronary arteries and pulmonary trunk, and vascular injury may cause ischemia and associated arrhythmias [97]. Generally, the occurrence of arrhythmias in cardiology is often centered around the pulmonary veins. A framework for radiotherapy planning was recently developed to define safe dosage limits for this specific structure [98] (Table 3). The analysis by Kim et al. showed that 3 year AF rates could be stratified by several cutpoints: SA node D_max_ ≥ 20 Gy vs. <20 Gy (10% vs. 1%, *p* < 0.001), RA D_max_ ≥ 19.1 Gy vs. <19.1 Gy (8% vs. 1%, *p* = 0.01), and LA D_max_ ≥ 55.8 Gy vs. <55.8 Gy (9% vs. 3%, *p* = 0.04) [95]. In clinical practice, comprehensive monitoring of effects of radiotherapy on the occurrence of cardiac side effects is required, including also its proarrhythmic effect. With a better understanding of RT-induced cardiac toxicity, as well as toxic dosimetric thresholds to the heart and its structures, modern radiotherapy techniques may be able to minimize cardiac toxicity while maximizing tumor control [99].

### 10.3. Chemotherapy and Targeted Therapies

Treatment for lung cancer depends on the cancer type, its stage, the presence of possible molecular mutations, and other factors. Table 1 shows the types of arrhythmias that may accompany the use of specific drugs in the treatment of lung cancer.

In patients treated with ICIs, the occurrence of arrhythmias is attributed to accompanying myocarditis, but they can also result from systemic inflammation, non-inflammatory left ventricular dysfunction, or local inflammation caused by myocardial NSCLC metastases [74].

There are few studies comparing the proarrhythmic potential of different chemotherapy regimens used to treat lung cancer. In a study by Chang et al. which included NSCLC patients, 3876 people treated with tyrosine kinase inhibitors (TKIs) were matched to 3876 patients treated with platinum analogues [101]. They found that the risk of AF was similar in both groups. However, the risk of both VA and SCD was significantly greater among TKIs-treated patients compared with those treated with platinum analogues. In addition, the increased risk of VA/SCD persisted regardless of sex and the presence of most cardiovascular diseases.

## 11. Directions

Cancers and heart diseases are currently the leading causes of death in our population today, underscoring the need for further development of cardio-oncology. The training of specialists in this field is essential to address the increasing number of cardiovascular complications being diagnosed.

Research on cardiac dosimetry and its impact on cardiovascular toxicity in lung cancer is ongoing. The identification of dose thresholds that are toxic to the heart and its structures, together with modern radiotherapy techniques and the identification of risk factors, may minimize cardiovascular toxicity and thus the incidence of cardiac arrhythmias. There is also some evidence that statins may reduce RT-induced vascular damage, but this needs to be confirmed in randomized trials.

Cardiotoxicity may occur both during anticancer treatment and at various times after its completion. Therefore, patients scheduled for radical treatment for lung cancer should be referred to a cardiologist before treatment and then followed by him after treatment as part of regular specialist care. Improving cardiac care for patients with lung cancer will contribute to improving treatment outcomes in this group of patients.

## 12. Conclusions

Arrhythmias in patients treated for lung cancer have not been extensively studied so far, apart from AF in patients undergoing surgery. Many factors may influence the occurrence of arrhythmias, but it cannot be ruled out that they may also result from the cardiotoxicity of anticancer treatment used for lung cancer, especially radiotherapy.

The most common arrhythmia in patients treated for lung cancer is AF. In antiarrhythmic management, the best choice is amiodarone, taking into account its dose reduction in the event of possible interactions with oncological drugs (mainly TKIs). There are reports of effective treatment of AF with CA, which also applies to other supraventricular arrhythmias as well as VT. In the prevention of thromboembolic complications in patients with AF, NOACs are increasingly used instead of LMWHs, which significantly improves effectiveness and patient comfort.

In patients with ventricular arrhythmias, amiodarone is also the best choice, as it is the strongest antiarrhythmic drug. In the case of recurrent VT, CA should be considered, and in patients with refractory VT despite CA, cardiac SBRT is a new promising treatment method.

There is growing evidence that radiotherapy used in the radical treatment of lung cancer has an adverse effect on the cardiovascular system, which may adversely affect survival. The vascular injury during RT may lead to acute ischemia. Another consequence of RT is myocardial fibrosis. In both cases, this may result in the development of a substrate for the occurrence of arrhythmias, both AF and malignant ventricular arrhythmias (VT or VF). 

## Figures and Tables

**Figure 1 cancers-15-05723-f001:**
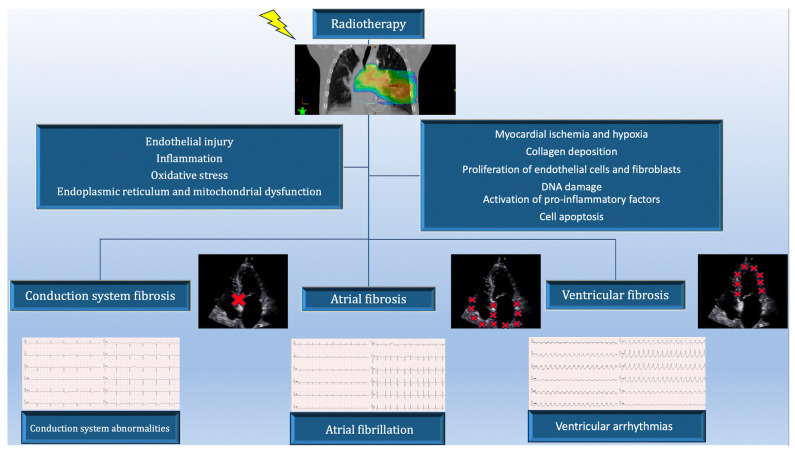
The mechanisms of radiation-induced cardiac arrhythmias. The upper part of the graphic-radiation dose distribution on a color scale in relation to heart structures. Red crosses mark regions of the heart potentially at risk of fibrosis, which may result in the cardiac arrhythmias presented below.

**Table 1 cancers-15-05723-t001:** Chemotherapeutic agents, targeted therapies, and immunotherapies used to treat lung cancer and associated cardiac arrhythmias.

Drug	Drug Class	Associated Arrhythmias
Doxorubicin	Anthracyclines	Sinus bradycardia, sinus tachycardia, supraventricular and ventricular arrhythmias (rarely reported life-threatening arrhythmias), AV block, other conduction disturbances [17]
Methotrexate	Antimetabolites	Sinus bradycardia, supraventricular tachycardia, VT, VF [15]
Vincristine	Vinca alkaloids	AF, PVCs (very rarely) [15]
Cyclophosphamide	Alkylating agents	Sinus bradycardia, PACs, PVCs, supraventricular tachycardia, AF, ventricular arrhythmias, AV block [15]
Cisplatin	Platinum compounds	Sinus bradycardia, supraventricular tachycardia, AF (>10%), PACs, PVCs, VT (1–10%) [15,18]
Paclitaxel	Taxanes	Sinus bradycardia, sinus tachycardia (>10%), AF (<1%), AV block, VT (<1%) [15]
Alectinib	Multitargeted TKIs	Sinus bradycardia (5.1–20%), mean QTc change of 5.3 ms, QTc > 500 ms (0.45%) [15,19]
Ceritinib	Multi-targeted TKIs	Sinus bradycardia (3%), PACs, QTc > 500 ms (0.33%) [15,20]
Crizotinib	Multi-targeted TKIs	Sinus bradycardia (5–69%), mean QTc change of 9–13.3 ms, QTc > 500 ms (1.3%) [15,21]
Osimertinib	Multi-targeted TKIs	VT [22,23]
Pembrolizumab	ICIs	Rare cases of sinus tachycardia, ventricular bigeminy, AF, SCD [15,24]
Nivolumab	ICIs	VT (<1%), AV block [15,25]
Atezolizumab	ICIs	Complete AV block, AF [26]
Ipilimumab	ICIs	Limited reports on AF and malignant ventricular arrhythmias [15]

Abbreviations: AF—atrial fibrillation; AV—atrioventricular; ICIs—immune checkpoint inhibitors; PACs—premature atrial complexes; PVCs—premature ventricular complexes; QTc—corrected QT interval; SCD—sudden cardiac death; TKIs—tyrosine kinase inhibitors; VF—ventricular fibrillation; VT—ventricular tachycardia.

**Table 2 cancers-15-05723-t002:** Scores used for assessing the risk of thromboembolic complications (**A**), and the risk of major bleeding during antithrombotic treatment (**B**) in cancer patients with atrial fibrillation (AF).

(A) CHA2DS2-VASc Score for Atrial Fibrillation	Score	(B) HAS-BLED Bleeding Risk Score	Score
For risk stratification of ischemic stroke and thromboembolism in patients with AF			

**Congestive heart failure**	1	**Hypertension**	1
Signs/symptoms of heart failure confirmed with objective evidence of cardiac dysfunction			
**Hypertension**	1	**Abnormal liver function**	1
Resting BP > 140/90 mmHg on at least 2 occasions or current antihypertensive pharmacologic treatment		Liver disease, bilirubin > 2× ULN with ASAT/ALAT/ALP > 3× ULN	
**Age 75 years or older**	2	**Abnormal liver function**	1
		Dialysis, kidney transplantation, creatinine ≥ 200 µmol/L (≥2.26 mg/dL)	
**Diabetes mellitus**	1	**Stroke**	1
Fasting glucose > 125 mg/dL or treatment with oral hypoglycemic agent and/or insulin			
**Stroke, TIA, or SE**	2	**Drugs**	1
Includes any history of cerebral ischemia		Concomitant use of antiplatelet/NSAIDs	
**Vascular disease**	1	**Labile INR**	1
Prior myocardial infarction (MI), peripheral arterial disease (PAD), or aortic plaque		<60% time in therapeutic INR range	
**Age 65 to 74 years**	1	**Bleeding**	1
		Previous major bleeding or predisposition to bleeding	
**Sex category (female)**	1	**Alcohol**	1
Female gender confers higher risk		≥8 units/week	
		**Elderly:** age greater than 65 years	1

	**Maximum 9**		**Maximum 9**
***Score:*** 0 low risk 1–2 moderate/intermediate risk ≥3 high risk **Score ≥ 2–recommend anticoagulation**		***Score:*** 0 Low risk for major bleeding 1–2 Moderate risk for major bleeding ≥3 High risk of major bleeding >5 Very high risk of bleeding **If a score is ≥3 a caution is needed and regular patient review is recommended** **Consider LAA closure in high risk patients (score > 4) with a life expectancy > 12 months**	

Abbreviations: BP—blood pressure; ALAT—alanine transaminase, ALP—alkaline phosphatase, ASAT—aspartate transaminase, INR—international normalized ratio, LAA—left atrial appendage, NSAID—non-steroidal anti-inflammatory drugs, SE—systemic embolism, TIA—transient ischaemic attack, ULN—upper limit of normal.

**Table 3 cancers-15-05723-t003:** Radiation doses to individual heart substructures associated with the risk of cardiac arrhythmias.

Study (Ref)	Findings
Atkins KM et al. (2022) [97]	V5Gy ≥ 60 cm^3^ for the left atrium associated with increased risk of AF
Atkins KM et al. (2022) [97]	V60Gy ≥ 0.03 cm^3^ for the right atrium associated with increased the risk of SVT
Atkins KM et al. (2022) [97]	V5Gy ≥ 1 cm^3^ for the left main coronary artery was a risk factor of VT occurrence
Atkins KM et al. (2022) [97]	V10Gy ≥ 1 cm^3^ for the left main coronary artery increased the risk of bradyarrhythmias
Kim K [95] et al. (2022)	The maximum dose delivered to the sinoatrial node increased the risk of atrial fibrillation
Wang K et al. (2017) [100]	Risk factors for arrhythmic events: heart V5, right atrium V60, heart V30

Abbreviations: AF—atrial fibrillation; SVT—supraventricular tachycardia; VT—ventricular tachycardia.

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
