# Peer review of "Cardiac Arrhythmias in Patients Treated for Lung Cancer: A Review"

_cancers, 2023, doi:10.3390/cancers15245723_

Round 1

Reviewer 1 Report

Comments and Suggestions for Authors

This comprehensive review of the arrhythmic risk in lung cancer patients includes the most important information about this topic, and is of good quality.

However, sometimes it is not clear if the authors are speaking about cardiac arrhythmias in cancer patiants in general or in the specific subgroup of lung cancer.  At Page 2, for instance, I’d change the first sentence  of the paragraph 2 as follow: Generally speaking, Supraventricular and ventricular arrhythmias, atrioventricular blocks and sinus bradycardia are among the less common side effects of anticancer therapy [15]. However, Some antineoplastic drugs are well known as having a direct or indirect arrhythmogenic potential.

I’d suggest other minor changes:

Page 3: “Cardiac complications are related not only to the cancer itself and its treatment, but also to comorbidities, other non-oncological drugs, and electrolyte disturbances resulting from frequent side effects, such as vomiting or diarrhea, accompanying oncological treatment. “ Comment: This point is correct, very important, and should be stressed, adding (here or below) the suggestion of close monitoring of the electrolytes (including Potassium, Magnesium and Calcium) and correcting any even mild abnormality. This is useful both in preventing and in treating paroximal AF (See: Ann Thorac Surg. 2005 Jan;79(1):117-26.JAMA Netw Open. 2022 Oct 3;5(10):e2237234. ; J Thorac Dis. 2023 Sep 28;15(9):4648-4656. )

Page 10: The pathophysiology of AF induced by radiotherapy is most likely based on inflammation and later fibrosis of the atrial wall involved in the the radiating field. Moreover, RT affects the autonomic system, leading to inappropriate sinus tachycardia and to reduced Heart rate variability. See for instance:  Int J Cardiol, 240 (2017), pp. 196-202; Left atrial fibrosis in atrial fibrillation: mechanisms, clinical evaluation and management. J Cell Mol Med. 2021;25:2764–75; Adv Radiat Oncol. 2016;1:106–14.  And the very recent Radiation-associated Arrhythmias: Putative Pathophysiological Mechanisms, Prevalence, Screening and Management Strategies

Another point which should be expanded is the opportunity to close the left atrial appendage to reduce the cardioembolic risk due to atrial fibrillation, whenever the systemic anticoagulation is challenging.

On the other hand, the sections about the interventional cardiology for ventricular arrhythmias might be shortened.

Author Response

This comprehensive review of the arrhythmic risk in lung cancer patients includes the most important information about this topic, and is of good quality.

However, sometimes it is not clear if the authors are speaking about cardiac arrhythmias in cancer patiants in general or in the specific subgroup of lung cancer.  At Page 2, for instance, I’d change the first sentence  of the paragraph 2 as follow: Generally speaking, Supraventricular and ventricular arrhythmias, atrioventricular blocks and sinus bradycardia are among the less common side effects of anticancer therapy [15]. However, Some antineoplastic drugs are well known as having a direct or indirect arrhythmogenic potential.

Authors' response: Thank you for your comment. We have changed the quoted paragraph according to the reviewer's suggestion.

I’d suggest other minor changes: 

Page 3: “Cardiac complications are related not only to the cancer itself and its treatment, but also to comorbidities, other non-oncological drugs, and electrolyte disturbances resulting from frequent side effects, such as vomiting or diarrhea, accompanying oncological treatment. “ Comment: This point is correct, very important, and should be stressed, adding (here or below) the suggestion of close monitoring of the electrolytes (including Potassium, Magnesium and Calcium) and correcting any even mild abnormality. This is useful both in preventing and in treating paroximal AF (See: Ann Thorac Surg. 2005 Jan;79(1):117-26.; JAMA Netw Open. 2022 Oct 3;5(10):e2237234. ; J Thorac Dis. 2023 Sep 28;15(9):4648-4656. )

Authors' response: Thank you for your comment. At the reviewer's suggestion, we have added the applicable information to the manuscript. We have also added relevant references.

Page 10: The pathophysiology of AF induced by radiotherapy is most likely based on inflammation and later fibrosis of the atrial wall involved in the the radiating field. Moreover, RT affects the autonomic system, leading to inappropriate sinus tachycardia and to reduced Heart rate variability. See for instance:  Int J Cardiol, 240 (2017), pp. 196-202; Left atrial fibrosis in atrial fibrillation: mechanisms, clinical evaluation and management. J Cell Mol Med. 2021;25:2764–75; Adv Radiat Oncol. 2016;1:106–14.  And the very recent Radiation-associated Arrhythmias: Putative Pathophysiological Mechanisms, Prevalence, Screening and Management Strategies

Authors' response: At the reviewer's suggestion, we have added the applicable information to the manuscript. We have also added relevant references.

Another point which should be expanded is the opportunity to close the left atrial appendage to reduce the cardioembolic risk due to atrial fibrillation, whenever the systemic anticoagulation is challenging.

Authors' response: Thank you for this comment. We have expanded the section on LAA occlusion and added appropriate references.

On the other hand, the sections about the interventional cardiology for ventricular arrhythmias might be shortened.

Authors' response: Thank you for this comment. This paragraph has been shortened.

Reviewer 2 Report

Comments and Suggestions for Authors

Dear Authors,

Cardiac Arrhythmias in Patients Treated for Lung Cancer by Hawryszko et al summarizes the importance of cardiac arrhythmias and this is unpleasant to patients with lung cancer, in whom anticancer therapy, especially radiotherapy, may contribute to the onset of cardiac arrhythmias. The observed relationship between cardiac dosimetry and radiotherapy-induced cardiotoxicity in lung cancer treatment could explain the increased mortality from cardiovascular causes in patients after chest irradiation.  However, this manuscript needs additional data to improve the quality of the manuscript to be published in Cancers as a review article.

Major Comments:

1.       The authors could show a pictorial to demonstrate the mechanism of radiation-induced cardiac arrhythmias.

2.       The authors  could show chest MRI pictures before and after radiation therapy will improve the quality of the manuscript.

Author Response

Dear Authors,

Cardiac Arrhythmias in Patients Treated for Lung Cancer by Hawryszko et al summarizes the importance of cardiac arrhythmias and this is unpleasant to patients with lung cancer, in whom anticancer therapy, especially radiotherapy, may contribute to the onset of cardiac arrhythmias. The observed relationship between cardiac dosimetry and radiotherapy-induced cardiotoxicity in lung cancer treatment could explain the increased mortality from cardiovascular causes in patients after chest irradiation.  However, this manuscript needs additional data to improve the quality of the manuscript to be published in Cancers as a review article.

Major Comments:

  1. The authors could show a pictorial to demonstrate the mechanism of radiation-induced cardiac arrhythmias (Figure 1).

Authors' response: Following the reviewer's suggestion, we have added a diagram illustrating the mechanisms responsible for the occurrence of cardiac arrhythmias after radiotherapy.

  1. The authors  could show chest MRI pictures before and after radiation therapy will improve the quality of the manuscript.

Authors' response: Thank you for this comment. Unfortunately, our center does not routinely perform MRI examinations before and after radiotherapy. We rely mainly on CT or PET-CT images. To increase the value of the manuscript, we decided to add echocardiographic recordings with LVEF assessment using the 3D method, which showed a significant deterioration of left ventricular systolic function in a patient with atrial fibrillation with a fast ventricular rate (examination before the initiation of radiotherapy for lung cancer and after completion of radiotherapy). The recordings will be added as Supplementary files.

Round 2

Reviewer 2 Report

Comments and Suggestions for Authors

Dear authors,

The manuscript quality is improved.